# Strengths and Weaknesses of Cell Synchronization Protocols Based on Inhibition of DNA Synthesis

**DOI:** 10.3390/ijms221910759

**Published:** 2021-10-05

**Authors:** Anna Ligasová, Karel Koberna

**Affiliations:** Institute of Molecular and Translational Medicine, Faculty of Medicine and Dentistry and Czech Advanced Technology and Research Institute, Palacký University Olomouc, Hněvotínská 5, 779 00 Olomouc, Czech Republic

**Keywords:** DNA replication, cell cycle, S phase, thymidylate synthase, deoxyribonucleotide triphosphates synthesis, ribonucleotide reductase, thymidine

## Abstract

Synchronous cell populations are commonly used for the analysis of various aspects of cellular metabolism at specific stages of the cell cycle. Cell synchronization at a chosen cell cycle stage is most frequently achieved by inhibition of specific metabolic pathway(s). In this respect, various protocols have been developed to synchronize cells in particular cell cycle stages. In this review, we provide an overview of the protocols for cell synchronization of mammalian cells based on the inhibition of synthesis of DNA building blocks—deoxynucleotides and/or inhibition of DNA synthesis. The mechanism of action, examples of their use, and advantages and disadvantages are described with the aim of providing a guide for the selection of suitable protocol for different studied situations.

## 1. Introduction

Cellular growth and the preparation of cells for division between two successive cell divisions is known as the cell cycle. In eukaryotic cells, it includes two basic parts—interphase and the M phase. Interphase, a part of the cell cycle when cells are duplicating the genetic information, expressing proteins and growing, is further divided into three separate phases—G1 (gap 1), S (synthetic) and G2 (gap 2). G1 and G2 are usually characterized by cell growth and high metabolic activity. The nuclear DNA is replicated during S phase. M phase is divided into six stages. The first five stages, prophase, prometaphase, metaphase, anaphase and telophase, are commonly known as mitosis. Mitosis involves nuclear division (daughter chromosomes are separated). The sixth stage of the M phase is known as cytokinesis and involves cytoplasmic division- (cell is divided into two daughter cells). (Figure 1) [1,2].

The mammalian cell cycle is controlled by a subfamily of cyclin-dependent kinases (CDKs), the activity of which is modulated by several activators (cyclins) and inhibitors (Ink4, and Cip and Kip inhibitors) [3]. Progression through the cell cycle is controlled at distinct checkpoints: the G0/G1 checkpoint (restriction point), G1 checkpoint (or G1–S checkpoint), intra-S phase checkpoint, G2 checkpoint (or G2–M checkpoint) and mitosis-associated spindle assembly checkpoint [4,5,6].

Cell cycle deregulation is a common feature of human cancer. Cancer cells frequently display unscheduled proliferation, genomic instability (with increased DNA mutations and chromosomal aberrations) and chromosomal instability (changes in chromosome number) [3]. This fact, together with the necessity to specifically address various aspects of cellular life during the cell cycle, has resulted in the development of many protocols providing cell populations enriched in cells at the specific stage of the cell cycle.

Basically, two different approaches to obtain such cell populations are used: (i) the treatment of asynchronous cell populations using special chemical agents, resulting in cell arrest at the specific phase of the cell cycle or (ii) mechanical isolation of cells at specific phase of the cell cycle. While the first group of methods suffers from the fact that the used treatments can result in unwanted effects on the cellular metabolism, the second group frequently does not provide a sufficient number of cells in the particular cell cycle phase or the synchrony of cell yield is low.

The first group includes methods based on the arrest of cells at the specific point in the G1 phase (e.g., by serum or amino acid starvation); methods based on the blockade of S phase (e.g., by thymidine or hydroxyurea); approaches based on the cell arrest in the M phase (e.g., by nocodazole) or at the G2/M border (e.g., RO-3306; Figure 1). The second group includes the isolation of mitotic cells mostly by mitotic shake off, the elutriation method or isolation of cells using flow cytometry and cell sorters.

Here we summarize the possibilities of cell synchronization at the G1/S boundary by substances impairing the deoxynucleotide metabolism, and consequently DNA replication, by aphidicolin—an inhibitor of DNA polymerase α, and by mimosine—a plant aminoacid exhibiting various effects on cell metabolism resulting in inhibition of DNA replication. A brief overview of other frequently-used methods for cell synchronization is included as well.

## 2. Metabolism of DNA Precursors

As the substances used for the synchronization of mammalian cells at the G1/S boundary frequently target the synthesis of deoxynucleotides, a short introduction of their metabolism in mammalian cells is given first. A complete overview of nucleotides’ metabolism is summarized, for example, in [7].

The deoxynucleotides are de novo generated from ribonucleotides at the level of ribonucleotide diphosphates (adenosine diphosphate—ADP; guanosine 5′-diphosphate—GDP; cytidine 5′-diphosphate—CDP, and uridine 5′-diphosphate—UDP) by reduction at the 2’ position of the ribose subunit. This cytoplasmic reaction is catalyzed by the enzyme ribonucleotide reductase (RNR, Figure 2). Its appearance during evolution was a prerequisite for the transition from the “RNA world”, where RNA sufficed for both catalysis and information transfer, to today’s interplay among DNA, RNA, and proteins [8]. A general overview of the occurrence, catalytic function, regulation, and evolution of RNRs is reviewed, for example, in [9].

RNR is heterotetramer composed of two R1 subunits and two R2 subunits (Figure 2). During the S phase, the activity of RNR is greatly increased, while in the G1 phase its activity is very low [9,10]. In this respect, R2 subunit transcription, but not R1 subunit transcription, is repressed during G1 [9,11]. During mitosis, R2 subunits are degraded [12]. In resting cells (in G0 phase when cells are metabolically active but do not proliferate [1]), the R2 subunit is not transcribed [13]. It was found that the quiescent cells contain a second radical-providing small subunit, termed p53R2 with the same function as the homologous R2 [14]. Some data indicate that in the case of DNA repair, p53R2 is transcriptionally activated by p53 and translocates to the nucleus [14,15]. There, it can substitute for R2 forming a highly active RNR [10,16]. Besides its possible role in DNA repair, it was found that the p53R2 subunit has an essential role in mitochondrial DNA replication [14,17,18,19].

The enzyme activity is tightly regulated by allosteric regulation which prevents excessive concentration of each dNTP (deoxyribonucleotide triphosphate). R1 subunits contain an activity site, a specificity site and a catalytic site [20,21]. The activity site regulates the overall activity of the enzyme by binding of ATP (adenosine 5′-triphosphate; increase in the overall activity) and dATP (deoxyadenosine 5′-triphosphate; decrease in the overall enzyme activity (Figure 2) [19,22,23]. The specificity site regulates the substrate specificity. This site binds dGTP (deoxyguanosine 5′-triphosphate), dTTP (deoxythymidine 5′-triphosphate), ATP and dATP [20]. This binding determines the substrate preference [8,10,19,24,25]. Binding of ATP and dATP at the specificity site facilitates both CDP and UDP binding at the catalytic site. Binding of dTTP at the specificity site allows GDP binding at the catalytic site and dGTP binding at the specificity site facilitates ADP binding at the catalytic site [20]. Importantly, when dTTP is bound at the specificity site, it inhibits the reduction of both CDP and UDP [26]. After the ribonucleotide 5′ diphosphate reduction to deoxyribonucleotide 5′-diphosphate (Figure 3), the nucleoside diphosphate kinase catalyzes the transfer of the terminal phosphate groups from 5′-triphosphate to 5′-diphosphate nucleotides [27].

dTMP (deoxythymidine 5′-monophosphate) is de novo synthesized by thymidylate synthase (TS) from dUMP (deoxyuridine 5′-monophosphate). dUMP is generated mainly by the enzyme dUTPase which hydrolyses dUTP (deoxyuridine 5′-triphosphate) to dUMP and pyrophosphate. This reaction provides the substrate for thymidylate synthase and concurrently eliminates dUTP from the DNA biosynthetic pathway [28]. TS catalyzes the reductive methylation of dUMP to dTMP using *N^5^,N^10^*-methylenetetrahydrofolate as the one-carbon methyl donor [29]. *N^5^,N^10^*-methylenetetrahydrofolate is oxidized during this reaction to dihydrofolate and has to be regenerated by dihydrofolate reductase (DHFR) and serine hydroxymethyltransferase (Figure 4) [30]. The second pathway of dTTP synthesis is a salvage pathway (Figure 4). In this case, thymidine is converted to dTMP by the enzyme thymidine kinase. The thymidine comes from intracellular nucleic acid degradation or from extracellular nucleosides circulating in the bloodstream [31].

## 3. Targeting the Deoxynucleotide Metabolism and Its Use for Cell Synchronization

### 3.1. Thymidine

A high concentration of thymidine (Figure 5) is frequently used for cell synchronization at the G1/S boundary. After the addition to the culture medium, thymidine enters the cells and is rapidly converted to dTTP through a salvage pathway and its concentration in cells dramatically increases [32]. The mechanism of the thymidine action is based on the allosteric regulation of RNR enzyme when elevated dTTP concentration causes imbalance in the dNTP pool and inhibits reduction of CDP to dCDP by RNR [26,33,34]. The thymidine block can be reversed either by the thymidine removal or by the addition of deoxycytidine [35,36]. Depletion of the nuclear dCTP pool after the increased dTTP concentration has been observed, for example, in Chinese hamster ovary (CHO) cells. Simultaneously, great increase in nuclear pools of dGTP and dATP was measured [37]. Similar data were also obtained in Molm-13 cells [32]. On the other hand, incubation of L929 mouse cells with 5 mM thymidine resulted only in an increase in the dTTP pool. The pools of dATP, dGTP and dCTP were all reduced [38]. This shows that the reaction of cells after thymidine treatment can vary substantially depending on the particular cell line.

Typically, the thymidine concentrations used for cell synchronization are equal to, or above, 2 mM (see for example in [34,39]). The incubation time should be little longer than the sum of the lengths of G2, M and G1 phases. As cells in S phase could not transit this phase without the thymidine block removal, the synchronization by one thymidine block provides two populations of cells. One portion of cells is at G1/S boundary, the second one is trapped throughout the S phase. Therefore, a second block is usually performed after the release of cells from the first block. The time between release and the onset of the second block should somewhat exceed the length of the S phase. A typical protocol for HeLa cells can be found in [39,40].

### 3.2. Hydroxyurea

Hydroxyurea or hydroxycarbamide (Figure 5; HU) was first synthesized over a century ago in 1869 [41]. It is primarily used as an antineoplastic and antiviral agent [42]. HU inhibits RNR by directly reducing the diferric tyrosyl radical center in the smaller R2 subunit via a one-electron transfer from the drug [42]. HU thus inhibits production of dNTPs (Figure 3), and subsequently, also DNA synthesis. Because of the reversibility of its action, HU has commonly been used for cell synchronization. Its action is easily reversed by changing of the growth medium for drug-free medium. The treatment of cells by HU results in a decrease in purine pools in mammalian cells. Concerning the pyrimidine pools, conflicting data are available [37,38,43,44]. The complicated, often reciprocal, changes in individual dNTP pools occurring in HU-treated mammalian cells may be due to the compensatory activities of the deoxyribonucleotide salvage pathways in the higher eukaryotes [45].

As HU treatment also results in trapping DNA synthesizing cells in the S phase, the HU treatment is typically combined with alternative synchronization protocols. One example is the protocol comprising isoleucine starvation followed by incubation with hydroxyurea [46]. In this case, cells are first incubated in a culture medium lacking isoleucine for a time corresponding to the sum of the G1, S, G2 and M phases of the particular cell line. According to Tobey and Crissman [46], large quantities of cells may be reversibly arrested in early G1 by cultivation in an isoleucine-deficient medium. It was also shown that these cells do not enter a state of gross biochemical imbalance [47]. Then, the medium containing both isoleucine and hydroxyurea should be added, followed by the incubation of cells for a time period slightly exceeding the G1 phase length. The cells are released from the G1/S boundary after exchanging the medium for one without HU [46].

Instead of isoleucine starvation, serum deprivation (starvation) can be used before HU treatment as well [48,49]. Although both methods based on isoleucine or serum starvation are efficient, they are not convenient for all cell lines and therefore, the preliminary tests are necessary.

### 3.3. Aminopterin and Methotrexate

Aminopterin and methotrexate are analogues of folic acid (Figure 5) [50]. They are potent inhibitors of dihydrofolate reductase [51,52]. Folic acid (vitamin B9) is not synthesized de novo by mammalian cells, therefore, it has to be obtained from food [53]. It is reduced by the action of dihydrofolate reductase either partially to the intermediate dihydrofolate (DHF) or completely to tetrahydrofolate (THF) [54]. THF is important for metabolism of thymidine, purines, glycine, methionine and choline [55]. Consequently, its lack results in the cessation of DNA replication. Methionine and choline are commonly present in the cell culture medium. To minimize the negative effects on processes other than DNA replication during cell synchronization by these two drugs, cell culture media also contain, besides aminopterin or methotrexate, hypoxanthine and glycine. This focuses the effect of THF depletion on the thymidine metabolism and consequently on the DNA synthesis [56]. In the case of methotrexate, this effect is further deepened by its inhibition of the thymidylate synthase [57]. The inhibitory effect of both antifolate drugs can be overcome either by the medium exchange for an antifolate-free medium or by the addition of thymidine [56]. The protocol for the synchronization of cells using aminopterin can be found, e.g., in the studies by Adams (1969) or Lindsay et al. (1970) [58,59], and the protocol for methotrexate-based synchronization is described, for example, in [60,61].

As antifolate drugs require the presence of additional substances in the growth media during synchronization and an additional synchronization step is required to obtain the highly synchronized cell population, their popularity as synchronization agents is very low. On the other hand, methotrexate is one of the most effective and extensively used drugs for treating many kinds of cancer or severe and resistant forms of autoimmune diseases [62].

### 3.4. 5-Fluorodeoxyuridine

5-fluorodeoxyuridine (FdU; Figure 5) is an analogue of thymidine. It is transported into the cell where it is converted to FdUMP (fluorodeoxyuridine 5′-monophosphate) by the salvage pathway enzyme thymidine kinase [63]. A binary complex between 5-FdUMP and *N^5^,N^10^*-methylenetetrahydrofolate irreversibly inhibits thymidylate synthase, and thus blocks de novo synthesis of dTMP and results in the accumulation of dUMP [64]. FdU causes intracellular nucleotide pool imbalance with the decreased dTTP and increased dUTP levels and cessation of the DNA synthesis [65]. On the other hand, dUTP and FdUTP (fluorodeoxyuridine 5′-triphosphate) can be incorporated into DNA instead of dTTP, therefore, their incorporation results in base excision repair and excision of these nucleotides from the DNA [64].

If cells are growing in a culture medium with FdU which is also supplemented with thymidine, they are able to synthesize dTMP using the thymidine kinase (through the salvage pathway) [63]. FdU-mediated inhibition of DNA synthesis can therefore be reversed by the addition of thymidine. The protocol of synchronization can be found in [66]. However, FdU and its derivative 5-fluorouracil has mainly been used in the treatment of various solid tumors [67,68,69] and its use for cell synchronization is very uncommon.

### 3.5. Aphidicolin

Aphidicolin is a tetracyclic diterpenoid, obtained from *Cephalosporium aphidicola* (Figure 5) [70]. Aphidicolin inhibits the growth of eukaryotic cells by inhibiting the activity of DNA polymerase α without interfering with the activities of DNA polymerase β and γ. The effect of aphidicolin on DNA polymerase α is reversible [71]. Cell synchronization with aphidicolin is simple as aphidicolin-treated cells are released from the G1/S boundary by medium exchange [72]. On the other hand, similar to other protocols based on the DNA synthesis inhibitors, a large portion of cells is trapped in the S phase after aphidicolin treatment [73]. In this respect, aphidicolin treatment is usually combined with an additional synchronization step, e.g., with a subsequent second aphidicolin treatment after incubation of cells in an aphidicolin-free medium [73], with the mitotic shake-off [74] or with thymidine block [75].

### 3.6. Mimosine

Mimosine [β-[N-(3-hydroxy-4-oxypyridyl)]-α-aminopropionic acid] (Figure 5) also blocks cells at the G1/S border. It seems that this block is mediated by several mechanisms. It has previously been suggested that mimosine can (i) alter deoxyribonucleotide metabolism by inhibition of ribonucleotide reductase [76,77]; (ii) inhibit initiation of DNA replication at replication origins [78]; (iii) attenuate serine hydroxymethyltransferase [79] or (iv) enhance the levels of p27Kip1 [80]. The chelation of iron seems to be one of the main modes of action of mimosine for cell cycle arrest [81]. The other possible mechanisms of mimosine action are reviewed in [81]. Although mimosine’s effect on DNA replication is not completely clear, it is frequently used for cell synchronization on the G1/S boundary [82]. Mimosine action is also frequently combined with additional synchronization protocols to increase the percentage of cell synchrony at the G1/S border. Examples are the protocol combining thymidine block and mimosine treatment [82] or the protocol based on nocodazole and mimosine treatment [83].

## 4. Effects of Synchronization on Cellular Metabolism

Methods of cell synchronization based on targeting DNA replication are frequently used as they are relatively cheap and easy to perform; however, they exhibit several unwanted effects on the cell metabolism. These methods commonly result in trapping a relatively high proportion of cells in the S phase and this portion of cells encounters the consequences of replication stress as their replication forks are stalled. Stalled forks usually result in the formation of single stranded DNA (ssDNA) as replicative helicase continues to unwind the parental DNA [84]. The persistence of ssDNA, bound by replication protein A (RPA), and adjacent to the stalled newly replicated double-stranded DNA, generates a signal for activation of the replication stress response: a primer–template junction [84,85]. This structure serves as a signaling platform to recruit a number of replication-stress response proteins, including the protein kinase ataxia-telangiectasia mutated (ATM) and Rad3-related (ATR) [84,86,87,88,89]. This response promotes fork stabilization and restart, while preventing progression through the cell cycle until DNA replication is completed. If stalled forks are not stabilized, or if they persist for an extended period, replication forks will collapse. This collapse can result in the formation of double-stranded DNA breaks [84].

It has been reported that the exposure of cells to hydroxyurea, aphidicolin or thymidine at concentrations commonly used to synchronize cell populations led to the phosphorylation of histone H2AX on Ser139 (induction of γH2AX) through the activation of ATM and ATR protein kinase [90,91,92]. DNA damage caused by hydroxyurea or aphidicolin treatment was also documented by Hammond and colleagues [93]. In addition, chromosomal aberrations were observed after the use of thymidine treatment [94]. Further, it was also shown that the synchronization using thymidine, mimosine or aphidicolin may lead to growth imbalance and can also induce imbalance in the expression of cell cycle regulatory proteins such as cyclins B1, A and E [95].

Importantly, there are some cell cycle-dependent processes which are not inhibited or synchronized when DNA replication is arrested by hydroxyurea. Examples are centrosome replication [96] and *RRM2* transcription [13].

Moreover, protocols based on the inhibition of deoxynucleotide synthesis inevitably result in imbalances in the nucleotide pools with various effects on cell metabolism. For example, in the case of FdU, impaired dTMP biosynthesis results in accelerated rates of genomic uracil incorporation [97,98] and DNA repair leading to the accumulation of DNA strand breaks [99,100]. In addition, it is supposed that FdUMP, phosphorylated by thymidylate kinase and nucleoside diphosphate kinase to its triphosphate form (FdUTP), can be incorporated into DNA and contributes to FdU-mediated toxicity [101]. Further, it is supposed that the incorporated FdUTP is recognized and excised by base excision repair machinery using the same mechanisms that remove genomic uracil [102]. Therefore, this method should not be used for studies focused on issues dealing with the metabolism of deoxynucleotides or base excision repair.

As the efficacy of particular protocols depends on the cell metabolism, chosen protocol should be experimentally verified and optimized for every cell line. In this respect, the overexpression of thymidylate synthase can result into resistance to FdU [63,103]. The overexpression of DHFR can contribute to the resistance of cells to methotrexate [104]. Moreover, mutation of CHO cells causing resistance to aphidicolin was described in [105].

These data clearly show that, although the synchronization protocols based on the inhibition of DNA replication are easy to perform and can provide high amount of synchronized cells, they also have many negative effects on cell metabolism. These effects must be taken into account when planning the experiment.

## 5. Methods Overview

An overview of frequently used synchronization protocols, also involving those providing cells in cell cycle phases other than at G1/S border, is summarized in the Table 1. Irrespective of the protocol selection, optimization involving, e.g., dose and timing, should precede the experiments as too short incubation can result in insufficient cell synchronization while too long incubation can result in an increase in unwanted effects on cell metabolism.

## Figures and Tables

**Figure 1 ijms-22-10759-f001:**
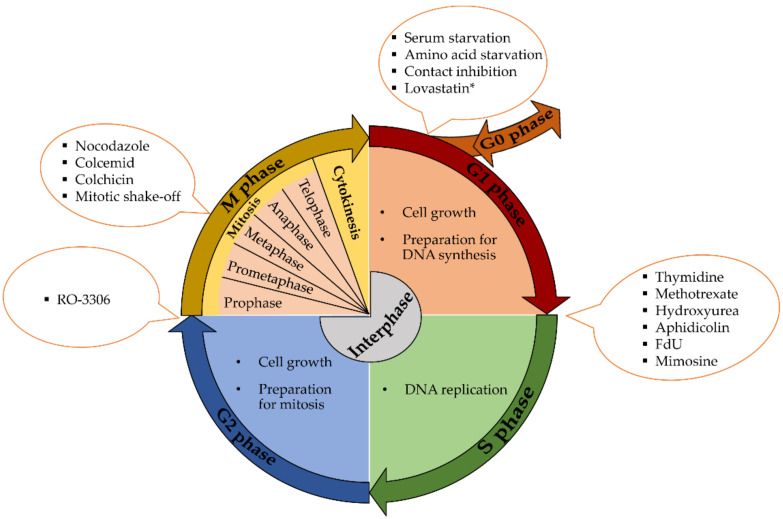
Overview of the cell cycle phases and some synchronization methods. * The stage of G1 phase at which lovastatin exerts its effect is not clear.

**Figure 2 ijms-22-10759-f002:**
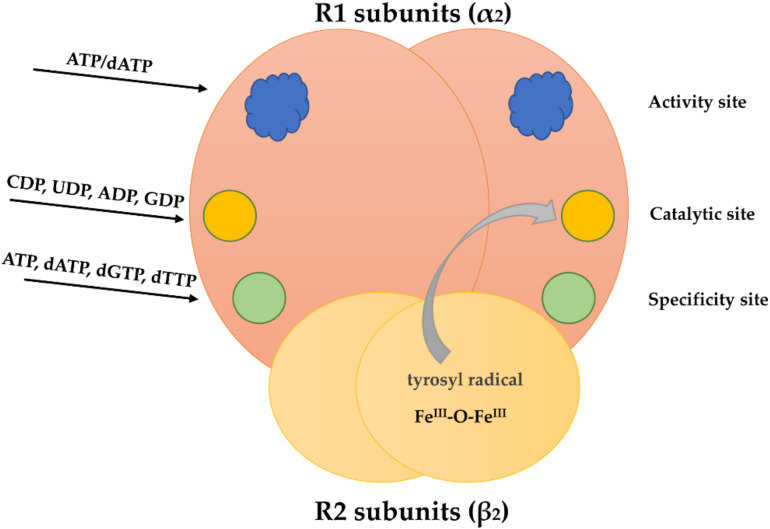
Schematic figure of the RNR heterotetramer. Each R1 subunit has two allosteric (activity and specificity sites) and one substrate binding site (catalytic site). The R2 subunits have a metal-oxygen center with a tyrosyl radical. This radical can be transferred to the catalytic site.

**Figure 3 ijms-22-10759-f003:**
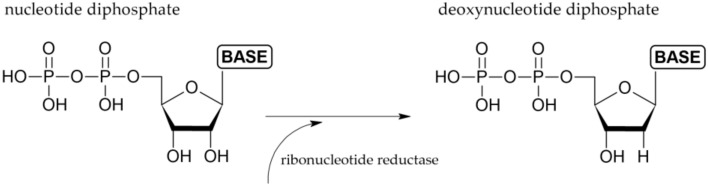
Simplified scheme of dNTP production.

**Figure 4 ijms-22-10759-f004:**
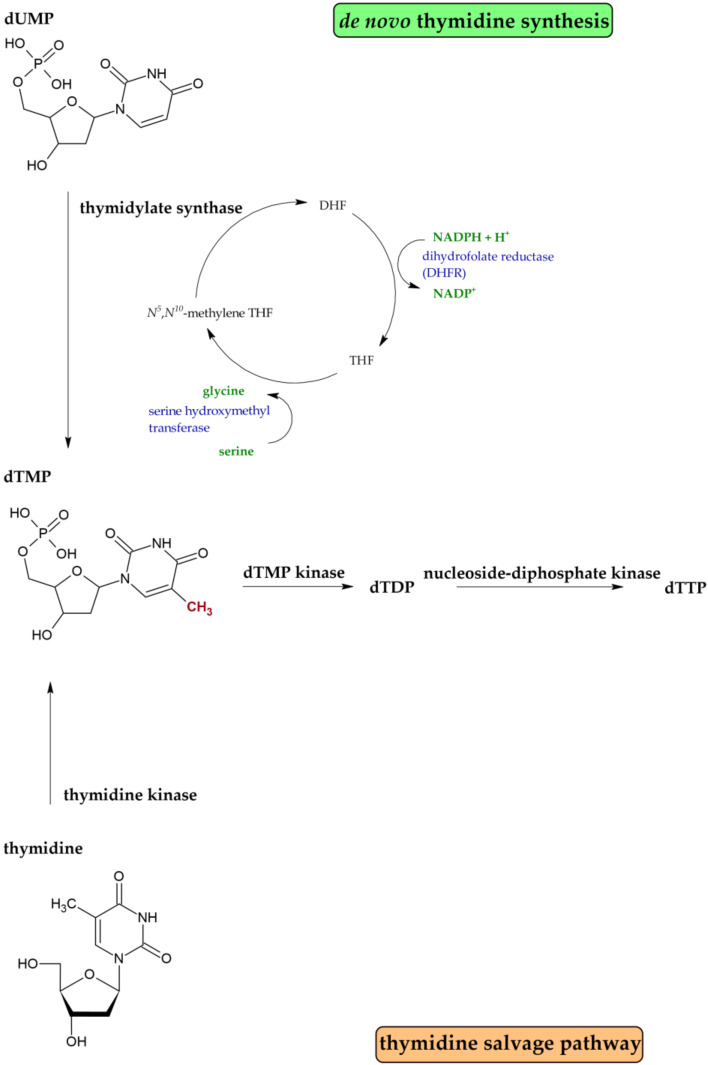
Simplified scheme of dTTP production by de novo synthesis or the salvage pathway.

**Figure 5 ijms-22-10759-f005:**
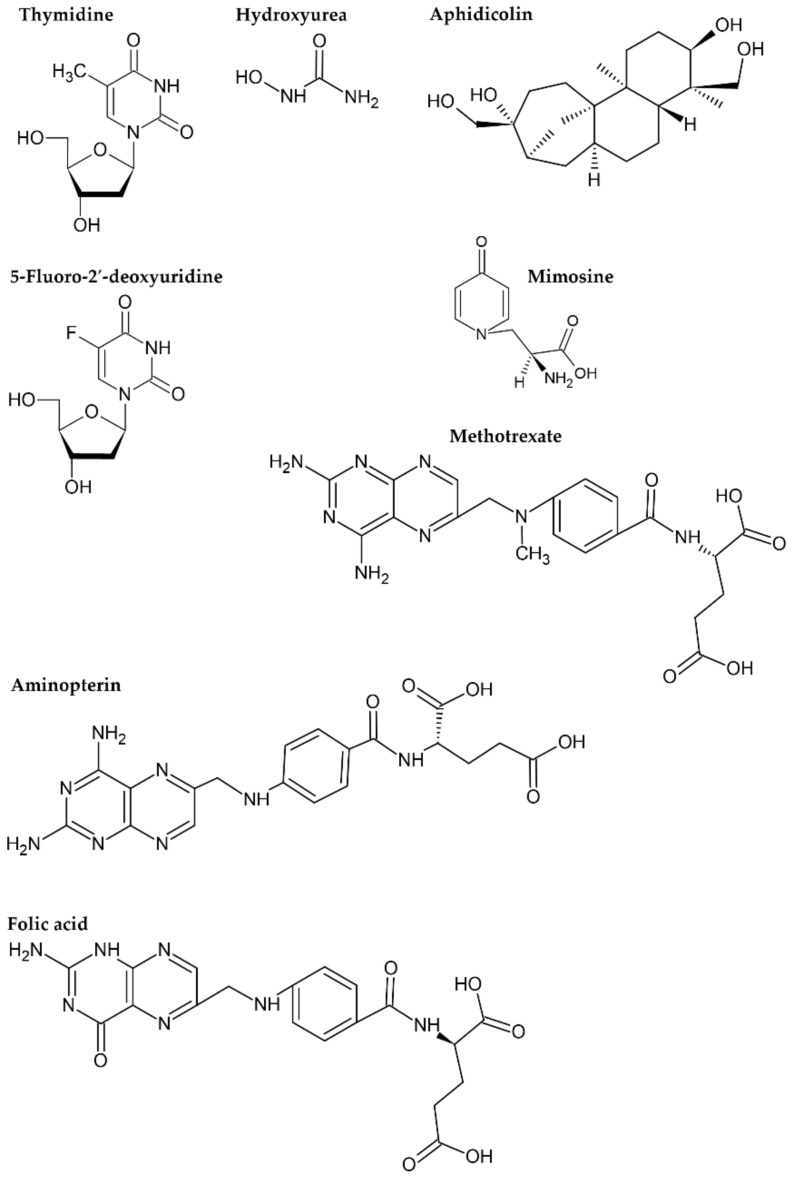
Formulae of the substances used for cell synchronization and of folic acid.

**Table 1 ijms-22-10759-t001:** Summarized overview of the commonly used synchronization approaches.

Method	Principle	Advantages	Disadvantages	Protocol
Mitotic shake-off	-Detachment of mitotic cells from cultivation surface by shaking or by medium flow	-Low effect on the cell metabolism -No special treatment necessary	-Low cell yield -Adherent cells only	[106]
Centrifugal elutriation	-Difference in sedimentation velocity (cell size dependence)	-Low effect on the cell metabolism -Preparation of G1, S and M fraction from one sample -No special treatment necessary	-Expensive instrument -Adjustment of convenient parameters is necessary -Adherent cells have to be released from the surface	[107]
Flow cytometry and cell sorting after DNA staining	-Differences in DNA content	-Preparation of G1, S a M fraction from one sample	-Impact of staining step with DNA dye on cell metabolism -Low cell yield -Adherent cells have to be released from the surface -Cell sorter required	[108]
Flow cytometry and cell sorting of unstained cells	-Differences in cell size	-No special treatment necessary	-Low resolution of cell cycle phases -Low cell yield -Cell sorter required -Adherent cells have to be released from the surface	[109]
Nocodazole/Colchicine/Colcemid	-Inhibition of mitotic spindle formation [110,111]	-Simple	-Prolonged treatment can result into aneuploidy, cell death or mitotic slippage [112]	[113,114,115]
RO-3306	-Inhibition of cyclin- dependent kinase 1 and consequently G2/M transition [116]	-Simple -Both adherent and suspension cell lines can be used	-Prolonged treatment can result into genome reduplication [117]	[118]
Lovastatin	-Mechanism is not completely understood -Lovastatin inhibits 3-hydroxy-3-methylglutaryl-coenzyme A reductase. -Decrease in the activity of cyclin-dependent kinase 1 was also documented [119].	-Simple -Both adherent and suspension cell lines can be used	-Can induce apoptosis [120] -The stage of G1-phase at which lovastatin exerts its effect is not clear.	[120]
Serum starvation	-Nutrient deprivation resulting into G0/G1 arrest [121]	-Simple -Cheap -Both adherent and suspension cell lines can be used	- Inappropriate for transformed cell lines -Prolonged serum starvation can result into DNA fragmentation [122]	[123]
Contact inhibition	-Contact inhibition of cell proliferation at high cell density resulting into G1 arrest [121,124]	-Simple -Cheap -Both adherent and suspension cell lines can be used	-Impropriate for non-adherent and transformed cells	[125]
Thymidine	-Inhibition of dCTP synthesis [26]	-Simple -Cheap -Both adherent and suspension cell lines can be used	-Induction of replication stress -Imbalance in nucleotide pools	[40,126]
Hydroxyurea	-Inhibition of dNTP synthesis [42]	-Simple -Both adherent and suspension cell lines can be used	-Induction of replication stress	[46,48,49]
Aphidicolin	-Inhibition of DNA polymerase α activity [71]	-Simple -Both adherent and suspension cell lines can be used	-Induction of replication stress	[73,74,75]
Mimosine	-Inhibition of RNR [77] -Inhibition of HMT [79] -Initiation of DNA synthesis at origins of replication [78]	-Simple -Both adherent and suspension cell lines can be used	-Induction of replication stress -Imbalance in nucleotide pools	[83]

## Data Availability

Not applicable.

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
