# Peer review of "Strengths and Weaknesses of Cell Synchronization Protocols Based on Inhibition of DNA Synthesis"

_ijms, 2021, doi:10.3390/ijms221910759_

Round 1

Reviewer 1 Report

This review by Ligasova and Koberna provides a nice summary of methods of cell synchronization based on inhibition of DNA synthesis. The metabolic processes are described such that the mode of action of the inhibitors can be understood. The advantages and disadvantages of different synchronization methods are discussed. It is generally well written and easy to understand. The figures are well made. I have only one concern. Figure 4 would be easier to follow if it was clear that N5,N10-methylene THF is being converted to DHF by thymidilyate synthase. As it is drawn the folate cycle is not connected with the remainder of the figure.

Author Response

Thank you for your valuable comments. Below find our replies.

  1. Figure 4 would be easier to follow if it was clear that N5,N10-methylene THF is being converted to DHF by thymidylate synthase. As it is drawn the folate cycle is not connected with the remainder of the figure.

We changed the Figure 4 (page 7). We hope that now it is easier to follow.

Reviewer 2 Report

In this manuscript, Ligasová and Koberna review various methods to synchronize cells in different phases of the cells cycle. They explain the biology behind each technique and discuss pros and cons of each approach. THe review mostly focuses on the synchronization protocols based on inhibition of DNA synthesis. It is clearly written review and will be useful for everybody studying the cell cycle, and I support its publication. I provide several suggestions that authors could use to clarify certain points. My comments are arranged by the order of the issues appearing in the text, and not by their importance.

  1. Page 1: “In M phase (mitosis), daughter chromosomes are separated and cell is divided into two daughter cells. It involves prophase, metaphase, anaphase and telophase (Figure 1) [1]”. Comment: Traditionally M Phase is divided into six stages (prometaphase and cytokinesis are missing). See e.g. Molecular Biology of the Cell, Alberts et al.
  2. Page 1: “The major checkpoints include the checkpoint at the entry into S phase (G1–S checkpoint), mitosis (G2–M checkpoint) and the spindle checkpoint that controls progression of cells into anaphase [3].” The question what is a checkpoint and what is a transition between two cell cycle phases is always confusing. I would at least call the first checkpoint a G1 or Restriction point or Start (triggered by serum / amino acid starvation), and not a G1-S checkpoint (as the authors correctly state it is triggered in G1). One could also mention the DNA damage / DNA replication checkpoint, as this is the checkpoint that arrests cells in S phase in response to dNTP depletion by HU and thymidine (as it is described in “4. Effects of synchronization on the cellular metabolism”).
  3. Page 2: “The deoxynucleotides are de novo generated from ribonucleotides at the level of ribonucleotide diphosphates”. Here and elsewhere I would stress that this review concerns primarily higher eukaryotes (mammalian cells) and RNRs belonging to class Ia. RNRs in class II (eukaryote Euglena gracilis and many prokaryotes) and class III (anaerobic RNRs) reduce triphosphates.
  4. Page 3: “In the case of DNA repair, p53R2 can substitute for R2 forming a highly active RNR [7,11].” It is now believed that p53R2 (more often called RRM2B) plays an essential role in mtDNA synthesis in non-dividing cells. It’s role in DNA repair is not well established. See https://pubmed.ncbi.nlm.nih.gov/24741716/

and https://www.nature.com/articles/ng0607-703

  1. Page 7: “It focus the effect of THF depletion on the thymidine metabolism and consequently on the DNA synthesis [49]” – unclear sentence. Do you mean “focuses”?
  2. Page 8: “It was previously suggested that mimosine can i) alter deoxyribonucleotide metabolism [69]; ii) inhibit initiation of DNA replication at replication origins [70]; iii) attenuate serine hydroxymethyltransferase [71]; iv) enhance the levels of p27Kip1 [72] or v) inhibit ribonucleotide reductase [73].” What is the difference between i) and v)?
  3. Page 9: “Indeed it was reported that the exposure of cells to hydroxyurea, aphidicolin or thymidine at concentrations commonly used to synchronize cell populations led to the phosphorylation of histone H2AX on Ser139 (induction of γH2AX) through the activation of ATM and ATR protein kinase [83-85].” In addition to chromosomal aberrations and various effects on cellular metabolism described below, the authors should probably consider that while hydroxyurea, aphidicolin or thymidine inhibit the chromosome cycle and cell division, certain processes in the cell cycle are not inhibited or synchronized when DNA replication is arrested. For example, centrosomes continue to duplicate in the cells when the chromosome cycle is inhibited by HU (https://journals.biologists.com/jcs/article/120/14/2444/29723/Centrosome-replication-in-hydroxyurea-arrested-CHO or https://pubmed.ncbi.nlm.nih.gov/17606999/). In another example, cell-cycle dependent transcription of the RRM2 gene (encoding the small subunit or RNR) is not affected by the cell cycle arrest by HU and has the same dynamic as in non-arrested cells  (https://pubmed.ncbi.nlm.nih.gov/10747958/). Thus, there are cell cycle programs that continue despite an arrested chromosome cycle.

Author Response

Thank you for all your valuable comments. Below find our replies.

  1. Page 1: “In M phase (mitosis), daughter chromosomes are separated and cell is divided into two daughter cells. It involves prophase, metaphase, anaphase and telophase (Figure 1) [1]”. Comment: Traditionally M Phase is divided into six stages (prometaphase and cytokinesis are missing). See e.g. Molecular Biology of the Cell, Alberts et al.

We rewrote the passage concerning the M phase (page 1, lines 24-32). We also changed Figure 1 accordingly (page 3).

  1. Page 1: “The major checkpoints include the checkpoint at the entry into S phase (G1–S checkpoint), mitosis (G2–M checkpoint) and the spindle checkpoint that controls progression of cells into anaphase [3].” The question what is a checkpoint and what is a transition between two cell cycle phases is always confusing. I would at least call the first checkpoint a G1 or Restriction point or Start (triggered by serum / amino acid starvation), and not a G1-S checkpoint (as the authors correctly state it is triggered in G1). One could also mention the DNA damage / DNA replication checkpoint, as this is the checkpoint that arrests cells in S phase in response to dNTP depletion by HU and thymidine (as it is described in “4. Effects of synchronization on the cellular metabolism”).

We rewrote the part that concerns checkpoints (page 1, lines 37-42).

  1. Page 2: “The deoxynucleotides are de novo generated from ribonucleotides at the level of ribonucleotide diphosphates”. Here and elsewhere I would stress that this review concerns primarily higher eukaryotes (mammalian cells) and RNRs belonging to class Ia. RNRs in class II (eukaryote Euglena gracilis and many prokaryotes) and class III (anaerobic RNRs) reduce triphosphates.

We added a note that the review is focused on the mammalian cells - Abstract (page 1, line 13) and chapter: „2. Metabolism of DNA precursors” (page 3, line 73 and page 4, line 75).

  1. Page 3: “In the case of DNA repair, p53R2 can substitute for R2 forming a highly active RNR [7,11].” It is now believed that p53R2 (more often called RRM2B) plays an essential role in mtDNA synthesis in non-dividing cells. It’s role in DNA repair is not well established. See https://pubmed.ncbi.nlm.nih.gov/24741716/ and https://www.nature.com/articles/ng0607-703

We rewrote the part concerning the p53R2 subunit and added the note that it is essential for mitochondrial DNA replication (page 4, lines 95-100).

  1. Page 7: “It focus the effect of THF depletion on the thymidine metabolism and consequently on the DNA synthesis [49]” – unclear sentence. Do you mean “focuses”?

We corrected the grammar mistake – in the sentence should be focuses (page 9, line 200).

  1. Page 8: “It was previously suggested that mimosine can i) alter deoxyribonucleotide metabolism [69]; ii) inhibit initiation of DNA replication at replication origins [70]; iii) attenuate serine hydroxymethyltransferase [71]; iv) enhance the levels of p27Kip1 [72] or v) inhibit ribonucleotide reductase [73].” What is the difference between i) and v)?

We combined these two points into one (page 10, lines 247-249).

  1. Page 9: “Indeed it was reported that the exposure of cells to hydroxyurea, aphidicolin or thymidine at concentrations commonly used to synchronize cell populations led to the phosphorylation of histone H2AX on Ser139 (induction of γH2AX) through the activation of ATM and ATR protein kinase [83-85].” In addition to chromosomal aberrations and various effects on cellular metabolism described below, the authors should probably consider that while hydroxyurea, aphidicolin or thymidine inhibit the chromosome cycle and cell division, certain processes in the cell cycle are not inhibited or synchronized when DNA replication is arrested. For example, centrosomes continue to duplicate in the cells when the chromosome cycle is inhibited by HU (https://journals.biologists.com/jcs/article/120/14/2444/29723/Centrosome-replication-in-hydroxyurea-arrested-CHO or https://pubmed.ncbi.nlm.nih.gov/17606999/). In another example, cell-cycle dependent transcription of the RRM2 gene (encoding the small subunit or RNR) is not affected by the cell cycle arrest by HU and has the same dynamic as in non-arrested cells (https://pubmed.ncbi.nlm.nih.gov/10747958/). Thus, there are cell cycle programs that continue despite an arrested chromosome cycle.

We added the note about it into the text (page 11, lines 283-285).